# DISTRIBUTION CALIBRATION FOR FEW-SHOT LEARNING BY BAYESIAN RELATION INFERENCE

## ABSTRACT

Learning from a limited number of samples is difficult as a small number of samples cannot cover all the information in their class. It is worth noting that classes with scarce samples may be distributed in a way that is related to classes that contain sufficient data. Therefore it is possible to calibrate the distribution of a sample-poor class by using classes with a large amount of data. Existing methods of distribution calibration usually use artificially set distances to calculate the association between two classes, which may ignore deeper relations between them. In this paper, we propose a distribution calibration method based on Bayesian relation inference. For the input few-sample classes, it can automatically infer their relation with the classes with sufficient data and adaptively generate a large amount of fusion feature data that can represent the few-sample classes. The results show that a simple logistic regression classifier trained by using the large amount of data generated by our method, exceeds state-of-the-art accuracy for skin disease classification issue. Through visual analysis, we demonstrate that the relation graphs generated by this Bayesian relation inference method have a degree of interpretability.

## 1 INTRODUCTION

In recent years, deep learning methods have been extremely successful in a variety of fields, such as image classification Lin et al. (2020); Yan et al. (2016), disease diagnosis Shen et al. (2019), thanks to the vast amount of available training data. In practice, however, the use of deep learning is greatly limited by the difficulty of obtaining a sufficient amount of training data for some tasks due to the scarcity of individuals and ethical issues Wang et al. (2020). This is most common in the medical field Zhu et al. (2020). In the case of dermatology, for example, classification of skin diseases from images is one of the key steps to the diagnosis of skin diseases. However, due to the diversity and complexity of diseases, it is difficult to obtain enough image data for many skin diseases. Even with human experts, the diagnosis is often subjective and inaccurate when there is insufficient data. Therefore, how to learn the right knowledge using a very small number of samples is an important area of research in artificial intelligence. There has been much work attempting to learn from a small number of samples of data, with early research typically using meta learning Finn et al. (2017) and metric learning algorithms Vinyals et al. (2016); Snell et al. (2017), which suffer from the difficulty of fully portraying class distributions with a small amount of data, resulting in lower accuracy. More advanced algorithms nowadays often use data augmentation methods to give the model better generalization abilities Zhou et al. (2021); Zhang et al. (2018). One of the most widely used methods is the distribution calibration of a small sample of classes by a known base class containing more data Yang et al. (2021b;a). However, most of the existing distribution calibration methods use artificially set metrics, such as the Euclidean distance between distributions Yang et al. (2021a), to perform similarity calculations between classes. While these methods yield relations between classes to a certain extent, such relations are often one-sided and it is difficult to capture potential relations between classes due to the complexity of the actual scenario (e.g. dermatological classification). In addition, it is difficult to fully characterise the relation between a small sample class and the base classes in a single relation strength judgement, as there is bound to be more or less variability between the classes.

To address the above challenges, we propose a few-shot learning method based on Bayesian relational inference, which can automatically infer potential distributional relations between classes and

generate a large number of Bayesian relation intensity graphs from multiple perspectives. Based on the large number of relation intensity graphs from different views fusing the characteristics of the base class distribution, the characteristics of the input target class are inscribed. Specifically, the main contribution points of the proposed model are summarized as follows:

- For the challenge of insufficient sample size, the model uses base classes containing more data to perform distribution calibration for a small sample class, generating a large number of feature embeddings that can represent small samples and alleviate the challenge of insufficient sample size.

- For the challenge that the current distribution calibration methods cannot infer potential relations between classes, we propose a Bayesian relation inference module to automatically infer potential relations between the small sample classes and the base classes without setting evaluation metrics manually. Besides, it and can make full use of the features of different base classes.

- For the challenge of large uncertainty in the relation between classes, we propose a multi-view Gaussian relation graph generation module to generate inferred graphs of the relationship between base classes and small sample classes from different angles, which can comprehensively portray the relations between classes.

## 2 RELATED WORKS

How to learn the right knowledge with very few samples is the goal of few-shot learning, and there has been many researches addressing this issue. According to the different methods used, few-shot learning is mainly divided into model-based fine-tuning, transfer-based learning and data augmentation. Fine-tuning is the most traditional method for few-shot learning. Early work was usually pre-trained on large datasets and later fine-tuned on few-shot datasets Howard & Ruder (2018); Naka-mura & Harada (2019). Some researchers use meta-learning methods to improve the fast adaptation of models Finn et al. (2017); Rusu et al. (2018); Nichol et al. (2018). Finn et al. use meta-learning methods to optimise the gradient descent procedure so that the model can solve new learning tasks with a small number of gradient descent steps and a small number of samples Finn et al. (2017). Some other researchers have proposed algorithms based on metric learning Bertinetto et al. (2018); Oreshkin et al. (2018), such as Matching Networks Vinyals et al. (2016) and Prototypical Networks Snell et al. (2017).

Data augmentation techniques have been widely used in advanced few-shot learning methods Chen et al. (2019); Yang et al. (2021a). Some researchers have introduced unlabelled data to improve model accuracy Ren et al. (2018); Liu et al. (2018). Ren et al. added unlabelled data to the Prototyp-ical Networks Snell et al. (2017) to improve model performance Ren et al. (2018). Some researchers have used contrast learning methods for few-shot learning. For example, Yang et al. Yang et al. (2022) used a combination of contrast learning and meta-learning for few-shot learning. These methods achieved better results, however they did not focus on the relations between classes.

Some researchers have attempted data augmentation by analysing the relations between classes. Yang et al. used the Euclidean distance between the mean and variance of the class distributions to infer the relationships between classes Yang et al. (2021a). These methods obtain relations between classes, however, they are unable to infer the underlying relations between classes, leading to limitations in the model.

To address these challenges, we propose a few-shot learning method based on Bayesian relational inference method that automatically inferred various relations between classes, generating enhanced fusion features that is more representative of the classes with insufficient data.

## 3 BAYESIAN DISTRIBUTION CALIBRATION

The overall framework of the proposed Bayesian distribution calibration method is shown in Figure 1. The data set is divided into four parts: base classes, training set, validation set and test set. The base classes are representative classes containing a large amount of data. The training process is a multi-classification task, which trains the Bayesian relation inference component to automatically

learn potential relations between classes. In the validation and testing process, we design a multi-view Gaussian graph generation component to generate a large number of relation graphs that can represent the potential relations between the target class and base classes. The final classification results are output by training a simple logistic regression classifier. Specifically, we put all the images in the dataset into the pre-trained Resnet18 feature extractor and use the generated features as feature embeddings for input to the Bayesian relational inference component. For the embeddings of base nodes, we use the mean of all embeddings in each class as node embeddings for relation inference. The node embeddings of base class nodes are calculated as shown below:

$$\boldsymbol{N_{b_i}} = mean(\sum_{X_{Y=y_i}} \boldsymbol{f}_{\mathrm{Res18}}(X_{Y=y_i})) \tag{1}$$

where $X_{Y=y_i}$ represents the input images labeled $y_i$, $y_i$ represents basenode classes, $\boldsymbol{f}_{\mathrm{Res18}}$ represents the pretrained Resnet18 model, and $\boldsymbol{N_b} \in \mathbb{R}^{\mathrm{K_b} \times \mathrm{K_f}}$ represents the embeddings of base class nodes. $mean(\cdot)$ stands for mean value calculation. Overall, The core ideas of our model are summarized as follows:

- Due to the scarcity of samples in some of the classes, we generate a sufficient number of fusion features by inferring the relations between the target class and base classes containing sufficient data.

- Due to the complexity of the relations between classes and the existence of many potential relations, we propose an automatic relation inference method based on the idea of VAE Kingma & Welling (2013). The method does not use a manually designed distance calculation method, and can adaptively infer the relations between target class and base classes, and generate inter-class relation intensity graphs.

- As the relations between classes are difficult to represent in a single relation graph, we propose a multi-view Gaussian graph generation method to generate multiple Gaussian relation graphs from different perspectives and generate data that can represent the distribution of the target class based on the Gaussian graphs from different views.

## 3.1 BAYESIAN RELATIONAL INFERENCE

How to make the model automatically learn potential relations between classes is challenging; there is some variation between classes as well as many similarities. In the case of skin diseases, for example, eczema and ringworm are divided into two classes due to their different etiologies, but both can cause itchy erythematous papules with a degree of similarity in images. Some diseases, although more different in image, may be very similar in terms of etiology, etc., and have a strong underlying association. Our proposed Bayesian relation inference method can effectively capture all types of relations between different classes, and its specific architecture is shown in Figure 2.

First, we generate edge embeddings from the input target class embedding and the node embeddings of the base classes to facilitate subsequent relation inference. Specifically, we stitch the target embedding to each base class embeddings separately and compute its edge embedding through a linear neural network. The edge embeddings are computed as shown below:

$$\boldsymbol{E}_{\mathrm{e}} = \boldsymbol{f}_\theta\left([\boldsymbol{N_t}, \boldsymbol{N_{b_i}}]\right) \quad (i \in [1, 2, \ldots, \mathrm{K_b}]) \tag{2}$$

where $\mathrm{K_b}$ represents the number of base class nodes, $\boldsymbol{N_t} \in \mathbb{R}^{\mathrm{K_f}}$ represents the target node embedding, $\boldsymbol{N_b} \in \mathbb{R}^{\mathrm{K_b} \times \mathrm{K_f}}$ represents the base node embeddings, $[\boldsymbol{N_t}, \boldsymbol{N_{b_i}}] \in \mathbb{R}^{2\mathrm{K_f}}$ represents the splicing of target node embedding and the $i^{th}$ base node embedding, $\boldsymbol{f}_\theta$ represents a linear neural network, and $\boldsymbol{E}_{\mathrm{e}} \in \mathbb{R}^{\mathrm{K_b} \times 2\mathrm{K_f}}$ represents the generated edge embeddings.

Coupling is one of the key steps in relational inference, and its purpose is to generate a summary graph of the relationship between the target class and the base class based on the edge embeddings. The relations between the classes are uncertain and complex. Besides, recent work mentions that humans engage in a myriad of unconscious perceptions when performing relation thinking which can be viewed as a sampling of a binomial distribution with $n \to \infty$ and $\lambda \to 0$ Huang et al. (2020). Hence we assume that the summary graph of the coupling is a sampling of a binomial distribution $\mathcal{B}(n, \lambda)$ with $n \to \infty$ and $\lambda \to 0$. However, due to the $n \to \infty$ of this binomial distribution, we cannot directly solve for the specific values of the edges of the summary graph. Based on De

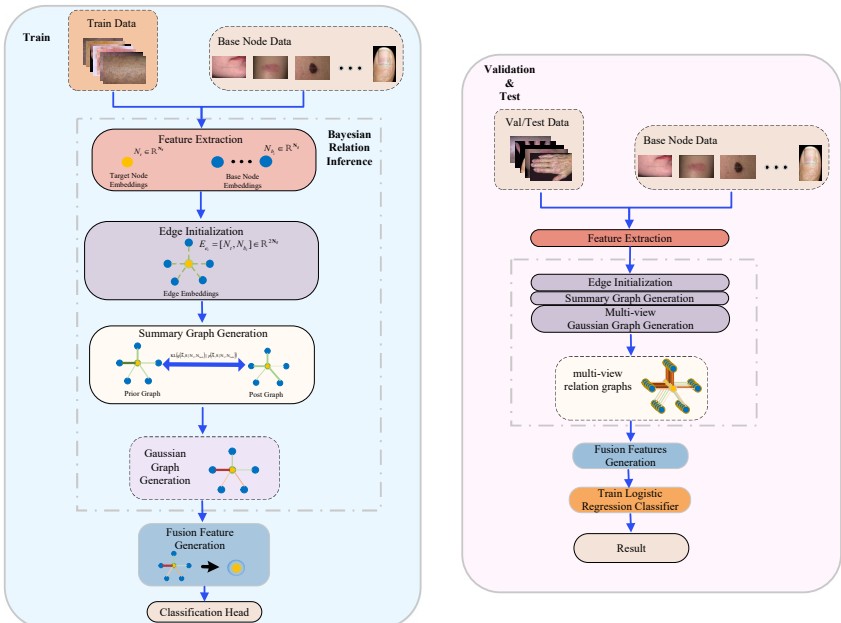

Figure 1: The overall architecture of the proposed Bayesian relation inference based distribution calibration for few-shot learning. With the Bayesian relation inference component, a summary graph of the relations between the target class and base classes can be obtained. Depending on the randomly generated Gaussian variables associated with the edges of the summary graph, a Gaussian graph that further represents the relations between the classes can be obtained from different views. In the training phase, we train a linear classification head, while in the validation and testing phases, we train a simple logistic regression classifier for few-shot classification tasks.

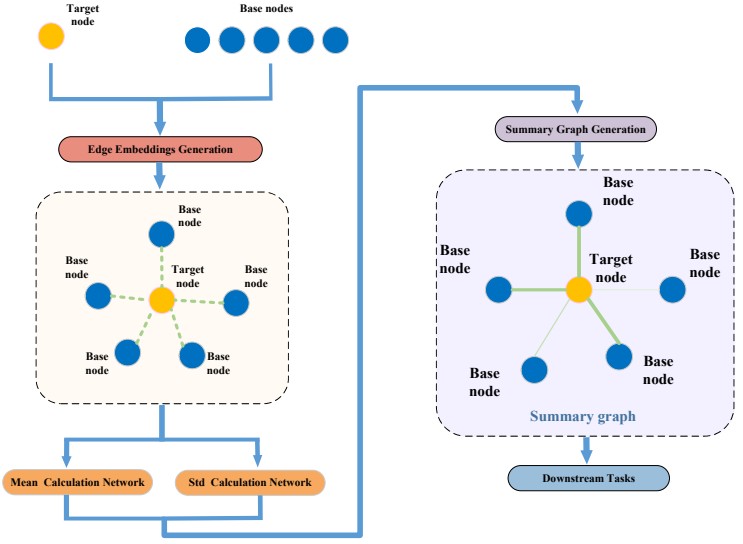

Figure 2: Detailed architecture of the Bayesian relation inference component. Inspired by VAE, we adopt two neural networks to fit the mean and variance of the summary graph, it can be then generated by these two figures. The summary graph can serve downstream tasks.

Moivre Laplace theorem and existing work Huang et al. (2020), we can convert this to a sampling

of a Gaussian distribution by the following conversion (Please refer to the supplementary material A.3.1 for the correctness of this conversion):

$$n_i = \zeta\left(\mathrm{L}_{\mathrm{mean}}\left(\boldsymbol{E}_{e_i}\right)\right) + \epsilon \tag{3}$$

$$\sigma_i = \zeta\left(\mathrm{L}_{\mathrm{std}}\left(\boldsymbol{E}_{e_i}\right)\right) \tag{4}$$

$$m_i = \frac{1 + 2n_i\,\sigma_i{}^2 - \sqrt{1 + 4n_i{}^2\,\sigma_i{}^4}}{2} \tag{5}$$

where $\zeta(\cdot)$ is softplus function, $\boldsymbol{E}_e \in \mathbb{R}^{K_b \times 2K_f}$ is the edge embedding generated in Eq.(2), $\epsilon$ is a very small constant, $\mathrm{L}_{\mathrm{mean}}(\cdot)$ and $\mathrm{L}_{\mathrm{std}}(\cdot)$ are implemented by neural networks for estimating the mean and standard deviation of the Gaussian distribution $\mathcal{N}\left(\mu_i,\ \sigma_i^2\right)$ which can calculate the approximation of the binomial distribution $\mathcal{B}(n, \lambda_i)$ with $n \to \infty$ and $\lambda_i \to 0$, and $\boldsymbol{M} \in \mathbb{R}^{K_b}$ is the summary graph which each $m_i \in \boldsymbol{M}$ is the sampling of the Gaussian distribution $\mathcal{N}\left(\mu_i,\ \sigma_i^2\right)$ and can strengthen the representation of the target class in relation to the base classes.

## 3.2 Multi-view Gaussian graphs and fusion feature generation

The summary graph can represent the relationship between the target class and the base class to some extent. However, in practice, the relations between classes are not invariant, and a single graph is a one-sided representation of the relations. In addition, due to the complexity of the relations between classes, the summary graph may be thickened by some unimportant relationships, which is not conducive to the construction of fusion features. In order to fully explore the important connections between classes and to generate enough sample data for downstream task, we propose a multi-view Gaussian graph generation method, which can generate a large number of sparse graphs that can represent the relations between classes from different views. Specifically, we introduce a large number of Gaussian variables associated with the edge weights of the summary graph to update its edge weights. For each view, the Gaussian graph is generated as follows:

$$\widetilde{\alpha}_i = \sqrt{(m_i \times (1.0 - m_i))} \times \varepsilon_i + m_i \tag{6}$$

$$s_i = m_i^{\mathrm{mean}} \times \widetilde{\alpha}_i + m_i^{\mathrm{std}} \times \sqrt{\widetilde{\alpha}_i} \times \varepsilon_i' \tag{7}$$

$$\bar{\alpha}_i = s_i \times \widetilde{\alpha}_i \tag{8}$$

$$\alpha_i = \zeta\left(\mathrm{L}\left(\bar{\alpha}_i\right)\right) \tag{9}$$

where $m_i \in \boldsymbol{M}$ is the approximation of the edges of the summary graph obtained in Eq.(5), $\varepsilon$ and $\varepsilon'$ are the standard Gaussian random variable of the same dimension as $\boldsymbol{M} \in \mathbb{R}^{K_b}$, $m_i^{\mathrm{mean}}$ and $m_i^{\mathrm{std}}$ represent the mean value and standard deviation of the Gaussian distribution $\mathcal{N}\left(\mu_i,\ \sigma_i^2\right)$ which can calculate the approximation of the binomial distribution mentioned in Section 3.1. By theorem 1 in supplementary material A.3.1, $\widetilde{\alpha}_i$ is a good approximation to this binomial distribution sampling. $\widetilde{\mathbf{A}} = \{\widetilde{\alpha}_1, \widetilde{\alpha}_2, \ldots, \widetilde{\alpha}_{n_b}\}$ is the Gaussian graph representation, $\boldsymbol{S}$ is the task-related Gaussian variable, $\bar{\mathbf{A}} = \{\bar{\alpha}_1, \bar{\alpha}_2, \ldots, \bar{\alpha}_{n_b}\}$ is the Gaussian transformation graph, $\mathbf{A} = \{\alpha_1, \alpha_2, \ldots, \alpha_{n_b}\}$ is the Gaussian relation graph, $\zeta(\cdot)$ is softplus function, and $\mathrm{L}(\cdot)$ is linear function. For different views, $\varepsilon'$ are regenerated to form Gaussian graphs in diverse perspectives

$$\check{\mathbf{A}} = \{\mathbf{A}^1, \mathbf{A}^2, \mathbf{A}^3 \ldots, \mathbf{A}^{n_g}\} \tag{10}$$

where $n_g$ represents the number of Gaussian graphs, $\check{\mathbf{A}}$ represents the multi-view Gaussian graphs.

To prevent the size of the generated Gaussian graph edges from unnecessarily affecting the generated data, we used a normalisation method to change the size of all edges to between -1 and 1, where a negative number represents a negative correlation between the base class and the target class, and a positive number represents a positive correlation between the two classes. Finally we generate fusion features by multiplying the edge size with the corresponding base class as the calibration value of that base class for the target class. This is calculated as follows:

$$\mathbf{A}_i' = \frac{2 \times \mathbf{A}_i - \mathbf{A}_{\max} - \mathbf{A}_{\min}}{\mathbf{A}_{\max} - \mathbf{A}_{\min}} \tag{11}$$

$$\check{\mathbf{A}}' = \{\mathbf{A}'^1, \mathbf{A}'^2, \mathbf{A}'^3 \ldots, \mathbf{A}'^{n_g}\} \tag{12}$$

for each normalised Gaussian graph $\mathbf{A}'^i$, a fusion feature embedding of target class can be generated as follow:

$$\boldsymbol{E}_{\text{out}}^{\text{i}} = \frac{N_b \otimes \mathbf{A}'^i}{\sum_j |\mathbf{A}'^i_j|} \quad (i \in [1, 2, \ldots, n_g]) \tag{13}$$

where $N_b$ represents the embeddings of base class nodes, $\boldsymbol{E}_{\text{out}}$ represents the fusion feature embeddings which can represents the distribution of the target class. In the training phase, we only generated a Gaussian relation graph and the corresponding fusion features in order to facilitate model optimization:

$$\boldsymbol{E}_{\text{out}}^{\text{train}} = \frac{N_b \otimes \mathbf{A}'}{\sum_j |\mathbf{A}'_j|} \tag{14}$$

### 3.3 LEARNING OF BAYESIAN RELATIONAL INFERENCE

In order to train the Bayesian relational inference component more effectively, inspired by VRNN Chung et al. (2015), we used a graph variational inference method to train the model and use the evidence lower bound (ELBO) for joint learning and inference. We use two random variables need to be optimised to describe the same random process data. Specifically, we use a Gaussian graph as the posterior graph $q\left(\tilde{\mathbf{A}}, \mathbf{S} \mid \mathbf{N}_{\text{t}}, \mathbf{N}_{1:n_b}\right)$, where the random variables $\tilde{\mathbf{A}}$ and $S$ can describe the same stochastic process, and the mean and variance of the prior graph $p\left(\tilde{\mathbf{A}}, \mathbf{S} \mid \mathbf{N}_{\text{t}}, \mathbf{N}_{1:n_b}\right)$ we fit using a linear neural network separately. Our goal is to maximize the following ELBO:

$$\sum_{i=1}^{M} \left( \text{KL}\left( q\left(\tilde{\mathbf{A}}_i, \mathbf{S}_i \mid \mathbf{N}_{\text{t}}, \mathbf{N}_{1:n_b}\right) \| p\left(\tilde{\mathbf{A}}_i, \mathbf{S}_i \mid \mathbf{N}_{\text{t}}, \mathbf{N}_{1:n_b}\right) \right) \right) \tag{15}$$

where $M$ is the number of Gaussian relation graphs generated in one batch. $q\left(\tilde{\mathbf{A}}_i, \mathbf{S}_i \mid \mathbf{N}_{\text{t}}, \mathbf{N}_{1:n_b}\right)$ is the prior graph, $p\left(\tilde{\mathbf{A}}_i, \mathbf{S}_i \mid \mathbf{N}_{\text{t}}, \mathbf{N}_{1:n_b}\right)$ is the posterior graph, $\tilde{\mathbf{A}}$ is the Gaussian graph presentation, and $\mathbf{S}$ is the task-related Gaussian variable. Since each variable in $\mathbf{S}$ is affected by $\tilde{\mathbf{A}}$ in Eq.(6), we have $s_i \mid \tilde{\alpha}_i \sim \mathcal{N}\left(\tilde{\alpha}_i * \mu_i, \tilde{\alpha}_i * \sigma_i^2\right)$. Besides, each element in Gaussian graph is conditioned on the Binomial variable for the same edge of the summary graph. Hence The KL term can be further written as:

$$\sum_{\text{i}}^{\text{K}_{\text{b}}} \left\{ \text{KL}\left( \mathcal{B}(\,n, \lambda_{\text{i}}\,) \| \mathcal{B}\left( n, \lambda_{\text{i}}^{(0)} \right) \right) \right. $$
$$\left. + \mathbb{E}_{\tilde{\alpha}_{\text{i}}} \left[ \text{KL}\left( \mathcal{N}\left( \tilde{\alpha}_{\text{i}} * \mu_i, \ \tilde{\alpha}_{\text{i}} * \sigma_{\text{i}}^2 \right) \| \mathcal{N}\left( \tilde{\alpha}_{\text{i}} * \mu_i^{(0)}, \tilde{\alpha}_{\text{i}} * \sigma_{\text{i}}^{(0)\,2} \right) \right) \right] \right\} \tag{16}$$

The calculation of this KL term does not depend on the random variable $\mathbf{S}$. By this equation we avoid the need to find the KL term for a large number of multi-view Gaussian graphs, which greatly simplifies the calculation. In Eq.(16), obviously the second term can be calculated, while the first term is tough to calculate because $n \to \infty$. According to recent researches Liu & Jia (2023); Huang et al. (2020), we can convert it into an easy-to-solve value to approximate the calculation (Please refer to the supplementary material A.3.2 for the calculation of this value).

## 4 EXPERIMENTS

### 4.1 DATASET

We use the Dermnet dataset[1], which contains 23 broad classes of dermatology images and can be manually divided into more detailed classes. The images in the dataset are in JPEG format, consisting of 3 channels, i.e. RGB. The resolutions vary from image to image, and from class to class,

---

[1]https://dermnet.com

but overall these are not extremely high resolution imagery.The classes include acne, melanoma, Eczema, Seborrheic Keratoses, Tinea Ringworm, Bullous disease, Poison Ivy, Psoriasis, Vascular Tumors, etc. To accommodate the few-shot classification task, we preprocess the dataset as described in Supplementary Material A.2.

## 4.2 COMPARISON EXPERIMENTS

To validate the effectiveness of the proposed model, we compare it with different baseline methods for few-shot learning. The baseline methods we compare are shown in Supplementary Material A.1.1.

Table 1: Comparison of Bayesian distribution calibration(BDC) and baselines on Dermnet dataset

| Method | 5way1shot(%) | 5way5shot(%) |
|---|---|---|
| MAML | 44.05 | 60.17 |
| PN | 43.76 | 60.22 |
| MN | 44.23 | 61.13 |
| DC | 48.99 | 66.75 |
| tSF | 49.38 | 68.15 |
| GAP | 48.92 | 68.89 |
| **BDC(Ours)** | **50.59** | **70.03** |

For a fair and meaningful comparison, we use the same setting to conduct comparisons with state-of-the-arts (SOTAs) on the Dermnet dataset. For all methods, we use Resnet18 pretrained model as the feature backbone and the dataset is divided in the same way as our method is set up. The experimental results show that our model achieves state-of-the-art performance at 5-way 1-shot and 5-way 5-shot experiment respectively. Traditional methods such as MAML, MN and PN have addressed the problem of few-shot learning tasks to a certain extent, but still suffer from problems such as lack of accuracy. Distribution Calibration(DC), a more advanced base-classes-based data augmentation method, uses the Euclidean distance between the mean and variance of features as a criterion for evaluating relations between classes, and has achieved greater success in few-shot classification tasks. However, as the manually set Euclidean distance has the disadvantage of not being able to evaluate the potential relations between classes, there is still some room for improvement in the classification accuracy of this method. Our model uses Bayesian relation inference methods to automatically infer relations between classes, allowing further improvements in classification accuracy. It can be seen that the classification accuracies obtained in 5-way 1-shot and 5-way 5-shot of our model are 50.59%, and 70.03%, respectively. Recent methods attempts to improve the performance of the model on few-shot classification tasks from different aspects, such as the introduction of transformer-based semantic filters Lai et al. (2022) and the Geometry-Adaptive Preconditioner Kang et al. (2023). These state-of-the-art methods substantially improve the performance of the model. However, they ignore the potential relations between the new class and the base classes, and our model outperforms them.

## 4.3 ABLATION EXPERIMENTS

In order to demonstrate the role of the individual components involved in the proposed Bayesian distributional calibration approach, we conducted a number of ablation experiments to quantify their enhancements to model performance. Specifically, we designed three variants of Bayesian distributional calibration including:

- Self-distribution Calibration (SDC), which uses the self-attention method for distribution calibration without base classes.The detailed architecture of this model is shown in Supplementary Material A.5.
- DIstribution Calibration (DC), which introduces base classes to aid in distribution calibration.
- BDC using Summary Graph (BDC-S), which introduces Bayesian relation inference method and taking summary graphs as relation intensity graphs.

Table 2 shows the performance of variant models on the derm dataset for 5way-1shot and 5way-5shot classification tasks. Among all the models, the performance of SDC is the worst, even lower than that of the traditional few-shot learning methods. This is due to the fact that The feature distribution of a single sample itself is highly randomized. Even with extensive training, it is difficult for a model to learn the real distribution of its class from a single image. The performance of DC is greatly improved due to the introduction of base classes assistance for distribution calibration of the input features. However, using the traditional Euclidean distance still cannot fully represent the association between base classes and target class. BDC-S generates more robust relation intensity graphs due to the introduction of the Bayesian relation inference method. The proposed model (BDC) introduces the Gaussian graph transform method based on the summary graph generated by BDC-S, and the obtained relation intensity graphs are more able to highlight the potential connections between classes, and the fusion features generated can better serve the downstream tasks, thus obtaining the best performance.

Table 2: Comparison of Bayesian distribution calibration(BDC) and baselines on Dermnet dataset

| Method | 5way1shot(%) | 5way5shot(%) |
|--------|--------------|--------------|
| SDC | 33.20 | 43.58 |
| DC | 48.99 | 66.75 |
| BDC-S | 49.76 | 67.96 |
| **BDC(Ours)** | **50.59** | **70.03** |

### 4.4 COMPARISON OF CONVENTIONAL ALGORITHMS AND FEW-SHOT LEARNING ALGORITHMS

In few-shot learning algorithms, training a model using conventional algorithms can be difficult due to the large number of categories and the fact that the data for most of the categories is scarce. We divide the test set of the Dermnet dataset in a ratio of 8:2 into a new training and test set. Then we generate a conventional algorithm model by replacing the few-shot classification part of the Bayesian relational inference model with a linear classification head. We freeze the Bayesian relation inference module and fine-tune the classification head on the new training set and test it on the new test set. We compare the accuracy of this algorithm with that of the few-shot learning algorithms on the 5-way 1-shot task. The experiment result is shown in Table 3.

Table 3: Comparison of few-shot algorithms and conventional algorithm on Dermnet dataset

| Method | Acc(%) |
|--------|--------|
| MAML | 44.05 |
| PN | 43.76 |
| MN | 44.23 |
| DC | 48.99 |
| tSF | 49.38 |
| GAP | 48.92 |
| *BDC + Conventional Algorthms* | *43.50* |
| **BDC(Ours)** | 50.59 |

The result shows that the conventional algorithm's accuracy is similar to that of the early few-shot algorithms on the 5-way 1-shot task. It is worth noting that traing the conventional algorithm is time-consuming and overall performs less well than the few-shot algorithms.

### 4.5 VISUALISATION ANALYSIS

To demonstrate the effectiveness of the relation inference graphs generated by our model, we visualise the relation graphs for different images, as shown in Figure 3. Specifically, we randomly select

images as target classes and take the average of the generated multi-view Gaussian graphs as the output. We visualise the strength of the relation between the target class and base classes in the form of a heat map, and select three base classes with strong positive correlations and three base classes with strong negative correlations for visual comparison with the target class image respectively.

The target class picture 1 is *Perioral Steroid*, which belongs to the *Acne and Rosacea Photos* major class in the Dermnet dataset. It can be seen that the onset of the disease in the image is mainly diffuse inflammatory response, which is similar to the symptoms of the base classes with a strong positive correlation, and it is noteworthy that the onset of the disease in target image 1 is on the lip, but the lip is not regarded as a basis for strong correlation in our relation inference model, but rather a more specialized onset symptoms as the main basis. In addition, it can be seen from the visualisation results that the onset symptoms of the three diseases with the strongest negative correlation with the target image 1 are clearly different from it. Meanwhile, the common onset sites of these three skin diseases are also different, which can demonstrate that the relations learned by the proposed model do not use the onset site as the main judgment indicator.

The target class picture 2 is *tufted-folliculitis*, which belongs to the *Hair Loss Photos Alopecia and other Hair Diseases* major class. It is worth noting that the target class image and the base class with which it has the strongest positive correlation are in the same major class in the Dermnet dataset, and it is also clear from the specific images that both disease classes result in some degree of hair loss and have a strong association. Correspondingly, the three base classes with the strongest negative correlations with the target images also show the differences between the classes more visually in the images.

Through visual analysis, we believe that the proposed model is interpretable and able to focus on deeper relations between different disease classes, facilitating its use in medical scenarios.

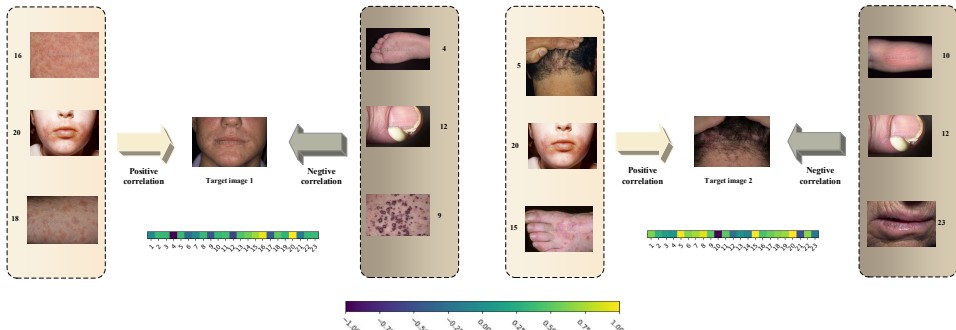

Figure 3: Visual analysis of the relationship graph generated by the proposed model. The heat map represents the association between the target image and the 23 base classes, with lighter colours representing positive correlations and darker colours representing negative correlations. The visual analysis shows that the proposed model is able to automatically infer the relation between the target disease and the base diseases based on the onset symptoms of the input images and adaptively generate relation graphs for better distribution calibration of target images.

## 5 CONCLUSION

We present a novel distribution calibration method based on Bayesian relation inference to solve the few-shot learning problem. To the best of our knowledge, this is the first attempt to combine the Bayesian relational inference method with distribution calibration. Specifically, we select some classes containing a large amount of data as the base classes, then infer the relation between the few-shot data and the base classes by Bayesian relation inference, and generate multi-view Gaussian relation graphs. The multi-view Gaussian relation graphs generated by the model are not only good at inferring potential relations between classes, but also solves the problem of uncertainty in the relations between them. With the multi-view relation graphs, we use the base classes to calibrate the distribution of the few-shot classes. We conduct extensive experiments on the Dermnet dataset and the experimental results show that our model achieves state-of-the-art results to date.

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
