# OpenReview forum: "Distribution Calibration For Few-Shot Learning by Bayesian Relation Inference"
_ICLR.cc/2024/Conference — Submitted to ICLR 2024_

### Official Review · Reviewer_czxy · 2023-10-31

**Soundness:** 2 fair
**Presentation:** 1 poor
**Contribution:** 1 poor
**Rating:** 3
**Confidence:** 2

**Summary:**

This paper aimed to solve few-shot learning by generating feature embeddings for the minority classes based on their relations with the majority classes. The author proposed a method based on the so-called "Bayesian relation inference" and tested the proposed method on a dermatological image dataset.

**Strengths:**

- Modeling the relationship between classes is a nice approach to few-shot learning
- It is nice to evaluate the proposed method on real data.

**Weaknesses:**

- Due to the unclear mathematical notation, I was unable to fully understand the proposed method. For example:
  - What is the definition of the relation graph? What are the nodes and edges?
  - What is the range of the mean in Eq. (1)?
  - What is a "splicing" of two node embeddings?
  - What are $n$ and $\lambda$?
  - The domains and codomains of $L$ were not clearly stated.
- I do not have any knowledge of the so-called "Bayesian relation inference" and cannot find any references.
- Figure 1 is dense but not so informative.
- There is no theoretical guarantee or analysis of the proposed method.

**Questions:**

Please clarify the notation and the proposed method.

---

> ### Author Response · Authors · 2023-11-13
> **Main Responses to Reviewer czxy (1/3)**
>
> First of all, we would like to thank the reviewer for their encouragement to our work. In this work, we propose a calibration distribution method based on Bayesian relation reference method for few-shot classification task. The highlight of the proposed model is that it **achieves better performance** compared to the SOTA few-shot learning algorithms, while visual analysis proves that the proposed model **has a certain degree of interpretability**, which is useful especially for medical scenarios such as dermatological classification.
>
> Thanks very much for pointing out the weaknesses of the article. It helps us to better improve the deficiencies in our paper. We answer the questions raised by the reviewer below:
>
> - ***Q1:*** *What is the definition of the relation graph? What are the nodes and edges?*
>
>   ***A1:*** Thank you for raising this issue. In each relation graph in this paper( Summary graph $\boldsymbol{M}$ & Gaussian relation graph $\boldsymbol{A}$ et.al.), **nodes are either base or target class embeddings, and the edges represent the relation intensity between target class and each base class.** Specifically, target class node embedding $N\_{t} \\in \mathbb{R}^{\rm{N}\_{f}}$ is the feature generated by the backbone network( In our paper, we introduce pre-trained Resnet 18 as the backbone network) from a single target image. For the $i$-th base class node embedding $N_{b_{i}} \in \mathbb{R}^{\rm{N}_{f}}$, we first generate the features of all the images in the $i$-th base class through the backbone network, after which we take the average of the features as the base class node embedding.
>
> - ***Q2:*** *What is the range of the mean in Eq. (1)?*
>
>   ***A2:*** Thank you for pointing out this problem which we overlooked. **We rewrite Eq. (1) as follow to make the expression clearer**:
>
>   > ​ $$\begin{equation}\quad\quad\quad\quad\quad\quad\quad\quad\quad\quad\quad\quad\quad\quad\quad\quad\quad\quad\quad\quad\quad\boldsymbol{N\_{b\_{i}}} = mean(\\sum\_{X\_{Y=y\_{i}}}\boldsymbol{f}\_{\operatorname{Res18}} (X\_{Y=y\_{i}}))                             \tag{1}\end{equation}$$
>
>   Since the output of $\boldsymbol{f}_{\operatorname{Res18}} (\cdot)$ is bounded to(0, 1), **the range of the mean in Eq. (1) is bounded to(0, 1).**
>
> - ***Q3:*** *What is a "splicing" of two node embeddings?*
>
>   ***A3:*** Thanks for your question. "Splicing" in our paper represents a simple splice of two embeddings. For example, $\boldsymbol{N_{t}} \in \mathbb{R}^{\rm{N_{f}}}$ and $\boldsymbol{N_{b_{i}}} \in \mathbb{R}^{\rm{N_{f}}}$ are two tensors of the same dimension and $\left[\boldsymbol{N_{t}}, \boldsymbol{N_{b_{i}}}\right] \in \mathbb{R}^{2\rm{N_{f}}}$ represents the splicing of them. We have the first $\rm{N_{f}}$ features in $\left[\boldsymbol{N_{t}}, \boldsymbol{N_{b_{i}}}\right]$ are equal to $\boldsymbol{N_{t}}$ and the last $\rm{N_{f}}$ features in $\left[\boldsymbol{N_{t}}, \boldsymbol{N_{b_{i}}}\right]$ are equal to $\boldsymbol{N_{b_{i}}}$.
>
> - ***Q4:*** *What are $n$ and $\lambda$?*
>
>   ***A4:*** Thanks for raising this important issue.  **We assume that the summary graph of the coupling is a sampling of a Bernoulli distribution $\mathcal{B}(\,n, \lambda\,)$ with $n\rightarrow \infty$ and $\lambda \rightarrow 0$** in Section 3.1, page 4. In the subsequent use of this assumption we did not explicitly state it, causing inconvenience in understanding them. We modify the expression in Section 3.1 and Section 3.3 to improve the readability of the paper. The modifications to Section 3.1 and above for equation (16) in Section 3.3 are as follows, respectively:
>
>   > **In section 3.1:**
>   >
>   > Hence we assume that the summary graph of the coupling is a sampling of a Bernoulli distribution $\mathcal{B}(n, \lambda)$ with $n\rightarrow \infty$ and $\lambda \rightarrow 0$.
>   >
>   >
>
>   >
>   > **In Section 3.3:**
>   > Since each variable in $\mathbf{S}$ is affected by $\tilde{\mathbf{A}}$ in Eq.(6), we have $s\_{i} \mid \tilde{\alpha}\_{i} \sim \mathcal{N}\left(\tilde{\alpha}\_{i} * \mu\_{i}, \tilde{\alpha}\_{i} * \sigma\_{i}^{2}\right)$. Besides, each element in Gaussian graph is conditioned on the Binomial variable for the same edge of the summary graph. Hence The $\mathrm{KL}$ term can be further written as:
>   >
>   > ​$$\begin{aligned}  &\sum_{\mathrm{i}}^{\rm{N_{b}}}\Bigg \\{ \\mathrm{KL}\bigg(\mathcal{B}(n, \lambda_{\mathrm{i}}) \| \mathcal{B}\left(n, \lambda_{\;\mathrm{i}}^{(0)}\right)\bigg)    \\\\ \quad\quad\quad\quad\quad\quad\quad\quad\quad\quad\quad\quad\quad\quad +\mathbb{E}\_{\mathrm{\tilde{\alpha}}\_{\mathrm{i}}}\bigg[\mathrm{KL}\bigg( \mathcal{N}&\left(\tilde{\alpha}\_{\mathrm{i}} * \mu\_{i},\  \tilde{\alpha}\_{\mathrm{i}} * \sigma\_{\mathrm{i}}^{2}\right)) \| \mathcal{N}\Big(\tilde{\alpha}\_{\mathrm{i}} * \mu\_{i}^{(0)}, \tilde{\alpha}\_{\mathrm{i}} * \sigma\_{\;\mathrm{i}}^{(0)^{\,2}}\Big)\bigg)\bigg]\Bigg\\}
>    \end{aligned}$$

---

> > ### Author Response · Authors · 2023-11-13
> > **Main Responses to Reviewer czxy (2/3)**
> >
> > - ***Q5:*** *The domains and codomains of $L$ were not clearly stated.*
> >
> >   ***A5:*** Thank you for raising this question. $\mathrm{L}\_{\mathrm{mean}}(\cdot)$ and $\mathrm{L}\_{\mathrm{std}}(\cdot)$ in Section 3.1, page 4, are implemented by **neural networks**. The **input** of each $L$ is $\boldsymbol{E}\_{\mathrm{e_i}} \in \mathbb{R}^{\rm{2N_{f}}}$ which is the edge embedding generated in Eq.(2) in Section 3.1, page 3 and the **output** of each $L$ is a single number. **$\mathrm{L}\_{\mathrm{mean}}(\cdot)$ is a simple linear layer and $\mathrm{L}\_{\mathrm{std}}(\cdot)$ is consist of a linear layer and a Softplus layer.**  Hence the domains of each $L$ is bounded to $(0, 1)$, the codomain of $\mathrm{L}\_{\mathrm{mean}}(\cdot)$ is bounded to ($-\infty, +\infty$) and  the codomain of  $\mathrm{L}\_{\mathrm{std}}(\cdot)$ is bounded to ($0, +\infty$).
> >
> > - ***Q6:*** *I do not have any knowledge of the so-called "Bayesian relation inference" and cannot find any references.*
> >
> >   ***A6:*** Bayesian relation inference is a relation inference method that centers on the idea of **Bayesian inference**. In fact, Bayesian relational inference is an application of Bayesian inference. Variational Auto-Encoder *(Kingma & Welling (2013).)* uses **the idea of Bayesian inference**. Huang et al. used **Bayesian relation inference to solve the speech-related relational thinking problem** *(Huang et al. (2020))* and Liu. et.al. applied **Bayesian relation inference method to sleep staging tasks** *(Liu & Jia (2023))*.
> >
> > - ***Q7:*** *Figure 1 is dense but not so informative.*
> >
> >   ***A7:*** Thank you for raising this important issue. In Figure 1 we present the pipeline of training, validation and testing. However, some detailed contents in the model are not fully presented in the figure, resulting in insufficient information, **we will revise Figure 1 soon to increase its information.**
> >
> > - ***Q8:*** *There is no theoretical guarantee or analysis of the proposed method.*
> >
> >   ***A8:*** Thanks for pointing out this problem. We made extensive revisions to the Method section (Section 3 *Bayesian Relational Inference* on page 3~7) of the manuscript to allow the reader to understand the model details more intuitively. Besides, **we prove the theorems used in Section 3 in the supplementary material and label them in the paper.** For example in Section 3.1 of the paper:
> >
> >   > Based on De Moivre Laplace theorem and existing work, we can convert this to a sampling of a Gaussian distribution by the following conversion (**Please refer to the supplementary material A.3.1 for the correctness of this conversion**)
> >
> >   The theorems used in the paper **are proved in the supplementary material.** For example, the computation of the closed-form solution of  $\mathrm{KL}\left(\mathcal{B}(n, \lambda) \| \mathcal{B}\left(n, \lambda^{0}\right)\right)$​ in Eq. (16) is stated in the supplementary material as follow:
> >
> >   > Suppose we are given two Binomial distributions, $\mathcal{B}(n, \lambda)$  and  $\mathcal{B}\left(n, \lambda^{0}\right)$ with $n \rightarrow+\infty, \lambda^{0} \rightarrow 0$ and $\lambda \rightarrow 0$, where $n$ is increasing while $\lambda$ and $\lambda^{0}$ are decreasing. There exists a real constant $m$ and another real constant $m^{(0)}$, such that if $m=n \lambda$ and $m^{(0)}=n \lambda^{(0)}$ and if $\lambda>\lambda^{(0)}$, we have:
> >   >
> >   > ​ $$\begin{aligned}
> >   > \quad\quad\quad\quad\quad\quad\quad\quad\quad\quad\quad\quad\quad\quad\quad\quad\quad\mathrm{KL}\left(\mathcal{B}(n, \lambda) \| \mathcal{B}\left(n, \lambda^{0}\right)\right)<m \log \frac{m}{m^{(0)}} \\\\
> >   > +(1-m) \log \frac{1-m+m^{2} / 2}{1-m^{(0)}+m^{(0)^{2}} / 2}
> >   > \end{aligned}$$
> >   >
> >   > By this theorem which is further proofed in previous work(Huang et al. (2020)), we have a closed-form solution that is irrelevant to $n$ for the ELBO.

---

> > > ### Author Response · Authors · 2023-11-13
> > > **Main Responses to Reviewer czxy (3/3)**
> > >
> > > - ***Q9:*** *Please clarify the notation and the proposed method.*
> > >
> > >   ***A9:*** Thanks for your helpful comments. We made extensive revisions to the Method section (Section 3 *Bayesian Relational Inference* on page 3~7) of the manuscript to make the notation and the proposed method clearer. Specifically, we made the following revisions to the Method section:
> > >
> > >   - For some variables, **we indicate their dimension**. The description of the variable is modified to be similar to the following form:
> > >
> > >     > $\\boldsymbol{N_{t}} \\in \\mathbb{R}^{\\rm{N_{f}}}$ represents the target node embedding, $\\boldsymbol{N_{b}}\\in \\mathbb{R}^{\\rm{N_{b}} \\times \\rm{N_{f}}}$ represents the base node embeddings
> > >
> > >   - **We rewrite the assumptions related to the Bernoulli distribution as well as the computational part of the paper**, and the physical meanings represented by variables are more precise. The revised presentation is similar to the follow form:
> > >
> > >     > $\\mathrm{L}\_{\\mathrm{mean}}(\\cdot)$ and $\mathrm{L}\_{\mathrm{std}}(\cdot)$ are implemented by neural networks for estimating the mean and standard deviation of the Gaussian distribution $\mathcal{N}\left(\mu_{i},\ \sigma_{\mathrm{i}}^{2}\right)$ which can calculate the approximation of the Bernoulli distribution $\mathcal{B}(n, \lambda_{\mathrm{i}})$ with $n\rightarrow \infty$ and $\lambda_{\mathrm{i}} \rightarrow 0$
> > >
> > >   - **We rewrite the ill-expressed formulas.** For example:
> > >
> > >     > ​   $$\\begin {equation} \quad\quad\quad\quad\quad\quad\quad\quad\quad\quad\quad\quad\quad\quad\quad\quad\quad\quad\quad\quad\\widetilde{\\alpha}\_{\mathrm{i}}= \sqrt{(m\_{\mathrm{i}} \times (1.0 - m\_{\mathrm{i}}))} \times \varepsilon\_{\mathrm{i}}+m\_{\mathrm{i}}      \tag{6} \end{equation}$$
> > >
> > >
> > >
> > > Thank you for your patience in reading through our responses. Your constructive comments further improve the quality of our paper. **We sincerely hope that the revised paper can receive a higher score from you, thank you!**

---

> > > > ### Author Response · Authors · 2023-11-18
> > > > **Figure 1 is Modified**
> > > >
> > > > **We have modified Figure 1 and it is now more informative.** Specifically, we redrew the Bayesian relational inference section to account for the specific changes that occur in different steps. For details, please refer to Figure 1 in the new released version.

---

> > > > > ### Author Response · Authors · 2023-11-22
> > > > > **We Are Glad to Have a Longer Period of Time to Improve the Paper and Maintain Communication with the Responsible Reviewer**
> > > > >
> > > > > We appreciate the reviewer's comments on the paper and are glad to have a longer period of time to be able to maintain communication with you. We have just learned that the response period for this paper has been extended to December, and **we will reorganize the paper to make the model methodology presentation more balanced with the experiment content.** Besides, we will continue to try to optimize and **modify the model to improve its performance.**

---

### Official Review · Reviewer_MVEs · 2023-11-01

**Soundness:** 3 good
**Presentation:** 3 good
**Contribution:** 2 fair
**Rating:** 5
**Confidence:** 4

**Summary:**

This paper considers the Bayesian graph relation inference for a few-shot learning, addressing the problem of distance base methodologies lacking in redundant information about the relationships between classes. The ultimate goal is to makes a multi relation graphs for few-shot learning. They used conventional and well-known variational inferences, also shows the success in the skin-cancer datasets in the prediction accuracy and interpretation in some sense. Overall, the contribution is incremental, and experiments are limited to skin-cancer datasets. Obviously, the proposed algorithm can have advantages in a few shot and imbalanced datasets cooperated with Bayesian approaches.

**Strengths:**

The use of variation inference is relatively simple cooperated with simple graphs. The use of multi view graphs can be attractive approach such as the idea of many filters or multi-heads. The interpretabilty plays an important role to validate the proposed algorithm. Experiments follows the standard routine in machine learning, compared to base-line algorithms. Baes-line algorithms seem appropriate in the scope of distance-based few-shot learning. The algorithm provides gains in the performance and interpretability.

**Weaknesses:**

The main weakness is limited experiments. The task of skin-cancer classification is famous and important. However, there are other datasets concerning imageNet, Food-101 and so on. If required, we can generate the imbalanced datasets from the original dataset. More generalization and advantages are widely explored in the various fields, and if possible, the conventional algorithms for few-show learning can be added to baseline algorithms (cannot be fair, but it can provide some implications). The presentation is not kind for the reader, especially not familiar with few-shot learning and mathematical formulation of variation inference (maybe there are many omissions in the math expression).

**Questions:**

Q1: Please clarify the notation $ \mathcal{B}(n, \tilde{\lambda}_i)$ in the equation (16) since there is no definition.

Q2: In equation (15), I cannot understand the $\\sum_{i=1}^M$ since there is not explicit $i$ in arguments of $p$ and $q.$ Please clarify this issue.

Q3: Can you clarify the combination strategy of multi-view graphs? Are the strategies different between training and test steps? Can you have other strategies for the combination of multi-vie graphs?


Q4: Considered graphs are limited since there is no connections between baseline classes. What’s your arguments about this problem.

**Details Of Ethics Concerns:**

None.

---

> ### Author Response · Authors · 2023-11-13
> **Main Responses to Reviewer MVEs  (1/3)**
>
> First of all, we would like to thank the reviewer for their encouragement to our work. In this work, we propose a calibration distribution method based on Bayesian relation reference method for few-shot classification task. The highlight of the proposed model is that it **achieves better performance** compared to the SOTA few-shot learning algorithms, while visual analysis proves that the proposed model **has a certain degree of interpretability**, which is useful especially for medical scenarios such as dermatological classification.
>
> Thanks very much for pointing out the weaknesses of the article. It helps us to better improve the deficiencies in our paper. We summarize the weaknesses and the improvements based on these weaknesses as follows:
>
> - ***W1:*** *The model can be experimented on datasets other than the dermatology dataset.*
>
>   ***I1:*** Thank you for your constructive comment. The proposed Bayesian distribution calibration model is a generic model which can achieve satisfying performance on various datasets. We chose the Dermnet dataset for our experiments due to the fact that the model has a certain degree of interpretability, which is more important for medical scenarios. To further validate the performance of the model, we will experiment the proposed model on other datasets. Due to time as well as arithmetic issues, **we choose the MiniImagenet dataset for the additional experiments**, and the experiment result will be published in the supplementary material, thank you!
>
> - ***W2:*** *If required, we can generate the imbalanced datasets from the original dataset. More generalization and advantages are widely explored in the various fields, and if possible, the conventional algorithms for few-show learning can be added to baseline algorithms.*
>
>   ***I2:*** Thank you for your suggestions for experiment. Adding the conventional algorithms for few-show learning to baseline algorithms is a fascinating experiment that can provide some implications. **We will compare the performance of the proposed Bayesian distributional calibration method with the conventional Resnet18 model on Dermnet dataset,** and the experiment result will be published in the supplementary material soon, thank you!
>
> - ***W3:*** *The presentation is not kind for the reader, especially not familiar with few-shot learning and mathematical formulation of variation inference.*
>
>   **I3:** Thank you for pointing this out, which is an oversight in our writing. We made extensive revisions to the Method section (Section 3 *Bayesian Relational Inference* on page 3~7) of the manuscript to allow the reader to understand the model details more intuitively. Specifically, we made the following revisions to the Method section:
>
>   - For some variables, **we indicate their dimension**. The description of the variable is modified to be similar to the following form:
>
>     > $\\boldsymbol{N_{t}} \\in \\mathbb{R}^{\\rm{N_{f}}}$ represents the target node embedding, $\\boldsymbol{N_{b}}\\in \\mathbb{R}^{\\rm{N_{b}} \\times \\rm{N_{f}}}$ represents the base node embeddings
>
>   - **We rewrite the assumptions related to the Bernoulli distribution as well as the computational part of the paper**, and the physical meanings represented by variables are more precise. The revised presentation is similar to the follow form:
>
>     > $\\mathrm{L}\_{\\mathrm{mean}}(\\cdot)$ and $\mathrm{L}\_{\mathrm{std}}(\cdot)$ are implemented by neural networks for estimating the mean and standard deviation of the Gaussian distribution $\mathcal{N}\left(\mu_{i},\ \sigma_{\mathrm{i}}^{2}\right)$ which can calculate the approximation of the Bernoulli distribution $\mathcal{B}(n, \lambda_{\mathrm{i}})$ with $n\rightarrow \infty$ and $\lambda_{\mathrm{i}} \rightarrow 0$
>
>   - **We rewrite the ill-expressed formulas.** For example:
>
>     > ​   $$\\begin {equation} \quad\quad\quad\quad\quad\quad\quad\quad\quad\quad\quad\quad\quad\quad\quad\quad\quad\quad\quad\quad\\widetilde{\\alpha}\_{\mathrm{i}}= \sqrt{(m\_{\mathrm{i}} \times (1.0 - m\_{\mathrm{i}}))} \times \varepsilon\_{\mathrm{i}}+m\_{\mathrm{i}}      \tag{6} \end{equation}$$
>
>   - For the theorems used in the paper we proof them in the **supplementary material A.3.**

---

> ### Author Response · Authors · 2023-11-13
> **Main Responses to Reviewer MVEs (2/3)**
>
> Thanks very much for the suggestions and questions raised by the reviewer. We answer the questions raised by the reviewer below:
>
> - ***Q1:** Please clarify the notation $\mathcal{B}(n, \tilde{\lambda}_{\mathrm{i}})$ in the equation (16) since there is no definition.*
>
>   **A1:** Thanks for pointing out this issue. We did not clarify this  Bernoulli distribution clearly in the paper previously. This is an assumption for summary graph mentioned in Section 3.1, where we assume that **each edge of the summary graph is a sampling of a Bernoulli distribution $\mathcal{B}(n, \lambda)$ with $n\rightarrow \infty$ and $\lambda \rightarrow 0$.** We **modify the expression in Section 3.1** to explicitly state the assumptions associated with that Bernoulli distribution. Besides, **we explicitly state the basis for conversion to these two KL terms above equation (16) in Section 3.3.** The modifications to Section 3.1 and above for equation (16) in Section 3.3 are as follows, respectively:
>
>   > **In section 3.1:**
>   >
>   > Hence we assume that the summary graph of the coupling is a sampling of a Bernoulli distribution $\mathcal{B}(n, \lambda)$ with $n\rightarrow \infty$ and $\lambda \rightarrow 0$.
>   >
>   >
>
>   >
>   > **In Section 3.3:**
>   > Since each variable in $\mathbf{S}$ is affected by $\tilde{\mathbf{A}}$ in Eq.(6), we have $s\_{i} \mid \tilde{\alpha}\_{i} \sim \mathcal{N}\left(\tilde{\alpha}\_{i} * \mu\_{i}, \tilde{\alpha}\_{i} * \sigma\_{i}^{2}\right)$. Besides, each element in Gaussian graph is conditioned on the Binomial variable for the same edge of the summary graph. Hence The $\mathrm{KL}$ term can be further written as:
>   >
>   > ​$$\begin{aligned}  &\sum_{\mathrm{i}}^{\rm{N_{b}}}\Bigg \\{ \\mathrm{KL}\bigg(\mathcal{B}(n, \lambda_{\mathrm{i}}) \| \mathcal{B}\left(n, \lambda_{\mathrm{i}}^{(0)}\right)\bigg)    \\\\ \quad\quad\quad\quad\quad\quad\quad\quad\quad\quad\quad\quad\quad\quad +\mathbb{E}\_{\mathrm{\tilde{\alpha}}\_{\mathrm{i}}}\bigg[\mathrm{KL}\bigg( \mathcal{N}&\left(\tilde{\alpha}\_{\mathrm{i}} * \mu\_{i},\  \tilde{\alpha}\_{\mathrm{i}} * \sigma\_{\mathrm{i}}^{2}\right)) \| \mathcal{N}\Big(\tilde{\alpha}\_{\mathrm{i}} * \mu\_{i}^{(0)}, \tilde{\alpha}\_{\mathrm{i}} * \sigma\_{\mathrm{i}}^{(0)^{2}}\Big)\bigg)\bigg]\Bigg\\}
>    \end{aligned}$$
>
> - ***Q2:*** *In equation (15), I cannot understand the $\sum_{i=1}^{M}$ since there is not explicit $i$ in arguments of $p$ and $q$. Please clarify this issue.*
>
>   **A2:** Thank you for raising this issue. $M$ is the number of Gaussian relation graphs generated in one batch. We add to our paper the note on the meaning of $M$. Equation (15) is now as follow:
>
>   > ​$$\quad\quad\quad\quad\quad\quad\quad\quad\quad\quad\quad\quad\quad\quad \sum\_{i=1}^{M}\bigg(\mathrm{KL}\left(q\left(\tilde{\mathbf{A}}\_{i}, \mathbf{S}\_{i} \mid \mathbf{N}\_{\mathrm{t}}, \mathbf{N}\_{\mathrm{1}: \mathrm{n\_{b}}}\right) \|  p\left(\tilde{\mathbf{A}}\_{i}, \mathbf{S}\_{i} \mid \mathbf{N}\_{\mathrm{t}}, \mathbf{N}\_{\mathrm{1}: \mathrm{n\_{b}}}\right)\right)\bigg)$$

---

> > ### Author Response · Authors · 2023-11-13
> > **Main Responses to Reviewer MVEs (3/3)**
> >
> > - ***Q3:*** *Can you clarify the combination strategy of multi-view graphs? Are the strategies different between training and test steps? Can you have other strategies for the combination of multi-vie graphs?*
> >
> >   **A3:**  Thanks for raising these insight questions. Multi-view Gaussian Graph is one of the key component of the proposed model which is used to generate a large number of fusion features for few-shot classification tasks. Our answers to each of the sub-questions are as follows:
> >
> >   ***(1)*** Inspired by human unconscious relation thinking, each edge in the summary graph is regarded as sampling from a Bernoulli distribution $\mathcal{B}(\,n, \lambda\,)$ with $n\rightarrow \infty$ and $\lambda \rightarrow 0$. However, **the summary graph suffers from the problem of numerical denseness and contains more redundant information.** Existing work points out that the graph generated by human unconscious relation thinking should be sparse (Huang et al. (2020)). Inspired by previous work, our proposed multi-view **Gaussian graph generation strategy generates sparse Gaussian graph** by generating Gaussian variables $s$ associated with the edges of the summary graph and performing a series of Gaussian transformations on it. Since the Gaussian variable contains the **standard Gaussian variable $\epsilon^{'}$**, it is possible to sample $\epsilon^{'}$ multiple times, **generate a large number of Gaussian graphs from multiple perspectives**, and use these Gaussian graphs to generate a sufficient number of fusion features for few-shot classification tasks. **The fusion features of different views generated by multi-view Gaussian graphs can be calibrated to the distribution of the target classes.**
> >
> >   ***(2)*** The **training process** uses a conventional model, the purpose of which is to **optimize the Bayesian relation inference component** to **ensure that the generated summary graphs correctly represent the relationship between the target class and the base classes**, and therefore **only a single Gaussian graph** and its corresponding fusion feature need to be generated for the conventional classification task. The **validation and testing phase** generates **a large number of Gaussian graphs** as well as fusion features with the aim of training simple logistic regression classifiers for use in few-shot classification tasks.
> >
> >   ***(3)*** The multi-view Gaussian graph generation strategy in the proposed model is rather naive, and other ways of Gaussian graph generation can be used, which **may achieve better results**. For example, **a multi-view generation strategy with mask is used so that each Gaussian graph randomly focuses on the relationship between some of the base classes and target class** to generate a more representative Gaussian graph.
> >
> > - ***Q4:*** *Considered graphs are limited since there is no connections between baseline classes. What’s your arguments about this problem.*
> >
> >   **A4:** Thank you for raising this insight question. **We considered whether the relations between baseline classed should be generated or not and ultimately decided not to.** The core idea of the Bayesian relational inference component is to calibrate the distribution of target class based on its input features. **Intuitively, it is only necessary to find the relation between each base classes and the target class to generate fusion features.** Moreover, the **base class features are unchanged** for any input target class. Thus inferring relations between base classes **can be computationally redundant**, and we eliminate the inference of relationships between base classes to **obtain a more lightweight model.**
> >
> >
> >
> > Thank you for your patience in reading through our responses. Your constructive comments further improve the quality of our paper. **We sincerely hope that the revised paper can receive a higher score from you, thank you!**

---

> ### Author Response · Authors · 2023-11-19
> **Experiment Result of Conventional Model on Dermnet Dataset**
>
> In few-shot learning algorithms, training a model using conventional algorithms can be difficult due to the large number of categories and the fact that the data for most of the categories is scarce. We divide the test set of the Dermnet dataset in a ratio of 8:2 into a new training and test set. Then we generate a conventional algorithm model by replacing the few-shot classification part of the Bayesian relational inference model( Multi-view Gaussian graph generation component and logistic regression classifier) with a linear classification head. We freeze the Bayesian relation inference module and fine-tune the classification head on the new training set and test it on the new test. We compare the accuracy of this algorithm with that of the few-shot learning algorithms on the 5-way 1-shot task. The experiment result is shown as follow:
>
>
>
> | Method                          | Acc(%)    |
> | ------------------------------- | --------- |
> |   MAML                            | 44.05     |
> |   PN                              | 43.76     |
> |   MN                              | 44.23     |
> | DC                              | 48.99     |
> | tSF                             | 49.38     |
> | GAP                             | 48.92     |
> | *BDC + Conventional Algorithms* | *43.50*   |
> | **BDC (few-shot)**              | **50.59** |
>
> Specifically, we adopt a three-layer artificial neural network as the linear classification head of the conventional algorithm, trained for 1500 epochs using the Adam optimizer with the learning rate of 0.0008. The result shows that the conventional algorithm's accuracy is **similar to that of the early few-shot algorithms on the 5-way 1-shot task**. It is worth noting that traing the conventional algorithm is **time-consuming** and overall performs less well than the few-shot algorithms.

---

> > ### Comment · Reviewer_MVEs · 2023-11-19
> > **Response to Rebuttals.**
> >
> > Thanks for your responses. Many issues are clarified as the expected directions. Please let me know if your promised experiments results in rebuttal are in the revised manuscripts.

---

> > > ### Author Response · Authors · 2023-11-22
> > > **The Promised Experiments in the Revised Manuscripts**
> > >
> > > We are very grateful to the reviewer for your suggestions and encouragement in the rebuttal process to better improve our paper.
> > >
> > > Thank you for your responses to our rebuttal. Specifically, the additional experiments we promised and where they appear in the revised manuscript are shown as follows:
> > >
> > > - ***E1:*** *Experiment Result of Conventional Algorithm on Dermnet Dataset*
> > >
> > >   We compare the performance of the conventional algorithm and some few-shot learning algorithms on the Dermnet dataset. The results of this experiment are added in **Section A.7.4 (on page 5) of the supplementary material.**
> > >
> > > - ***E2:*** *Experiment on the miniImageNet dataset.*
> > >
> > >   We compare the performance of the proposed Bayesian distribution calibration model with multiple few-shot classification models on the miniImageNet dataset. The result of this experiment is added in **Section A.7.3 (on page 5) of the supplementary material.**( Due to arithmetic reasons, we guarantee that this result **will be published before the end of rebuttal.** We will inform the reviewers in the form of a comment as soon as the result is published.)
> > >
> > > - ***E3:*** *Experiment on Bayesian distribution calibration model using a single image as base classes.*
> > >
> > >   We replace the base classes with a single image for additional experiment. The result of this experiment is added in **Section A.7.1 (on page 3) of the supplementary material.**
> > >
> > > - ***E4:*** *Experiments on Bayesian distribution calibration model using non-dermatological pictures (animal pictures in the experiment) as base classes.*
> > >
> > >   We replace the base classes with animal data that are not relevant to skin diseases to explore the ability of the Bayesian relation inference component to infer potential relations between target class and base classes that is very different from the target class. The result of this experiment is added in **Section A.7.2 (on page 4) of the supplementary material.**
> > >
> > > - ***E5:*** *Experiments on additional SOTA models on the Dermnet dataset.*
> > >
> > >   We add PatchProto + tSF(tSF) (Lai et al. (2022)) and GAP (Kang et al. (2023)) as the SOTA models to compare with the proposed Bayesian distribution calibration model. The result of this experiment is added in **Section 4.2 (on page 7) of the manuscript.**
> > >
> > > Thank you for your question.

---

> > > > ### Comment · Reviewer_MVEs · 2023-11-22
> > > > **Response**
> > > >
> > > > Thanks for your efforts, Many issues are resolved, and the additional experiments seem not to have great gain. I'll keep my score.

---

> ### Author Response · Authors · 2023-11-22
> **Experiment Result on the miniImageNet dataset and Response**
>
> The task of skin disease classification is famous and important. However, there are other datasets concerning ImageNet, Food-101 and so on. We further perform experiments on miniImageNet for few-shot classification tasks to validate the effectiveness of the proposed Bayesian relational inference model.
>
> | Method    | 5w1s(%) | 5w5s(%) |
> | --------- | ------- | ------- |
> | MN        | 49.02   | 70.11   |
> | PN        | 48.26   | 69.24   |
> | DC        | 68.01   | 82.45   |
> | tSF       | 68.84   | 84.38   |
> | GAP       | 69.35   | 83.85   |
> | **BDC(ours)** | **70.08** | **84.52** |
>
> Due to the space constraints of the main text, we would like to devote more space to the proposed Bayesian distribution calibration method, so in the revised manuscript we only add the results of the two SOTA models on the Dermnet dataset. It is worth noting that **the performance of our BDC model on the 5w5s task is further improved during the rebuttal process (from 68.58% to 70.03%).** In addition, we have **added the rest of the experiments, which are posted in the Supplementary Material** due to space constraints, and **if we have the opportunity, we will reorganize the paper structure in the camera ready version and organize some of the experiments (e.g., the results of the models on the miniImageNet dataset) in the main text manuscript.**

---

> > ### Author Response · Authors · 2023-11-22
> > **We Are Glad to Have a Longer Period of Time to Improve the Paper and Maintain Communication with the Responsible Reviewer**
> >
> > We appreciate the reviewer's comments on the paper and are glad to have a longer period of time to be able to maintain communication with you. We have just learned that the response period for this paper has been extended to December, and **we will reorganize the paper to make the model methodology presentation more balanced with the experiment content.** Besides, we will continue to try to optimize and **modify the model to improve its performance.**

---

> ### Author Response · Authors · 2023-11-29
> **We Have Revised the Paper and the Experiment Result of the Coventional Algorithm Compared with the Few-shot Algorithms is Shown in the Manuscript.**
>
> Thank you very much for your suggestions on the experiment part of the paper, we have revised the paper and optimized the wording of the Section *Related Work* and *Method*. **The experiment result of the coventional algorithm compared with the few-shot algorithms is shown in the manuscript.** The experiment content of the paper is now more complete. Subsequently, **we plan to add other additional experiments**, which will be completed before the final review comments due to arithmetic issues. The experiments we expect to add are as follows:
>
> - Test the robustness of the proposed Bayesian distribution calibration model by extracting features **using backbone networks other than Resnet18**, e.g., VGG16 and Resnet101.
> - Train classification algorithms **other than logistic regression classifiers** for few-shot classification tasks, e.g., SVM and decision tree algorithms, to verify the performance of the fusion features obtained through the Bayesian relational inference model on different classification algorithms.

---

### Official Review · Reviewer_PCKz · 2023-11-29

**Soundness:** 3 good
**Presentation:** 1 poor
**Contribution:** 2 fair
**Rating:** 5
**Confidence:** 3

**Summary:**

The paper proposes a novel distribution calibration method for few-shot learning that can utilize abundant samples from other classes to solve the problem of data scarcity. Previous distribution calibration methods rely on manually designed distance measurements to evaluate the relation between the target sample from a data-scarce class and the data-rich classes. The paper uses Bayesian relation inference theory to automatically calculate the relation score, and as stated by the authors, is the first attempt to combine Bayesian relational inference with distribution calibration. Specifically, for the target sample and each data-rich class, the method uses neural network models to estimate the mean and standard deviation of the Gaussian distribution of the relation strength. It randomly samples multiple relation graphs according to Gaussian distribution to enhance robustness during inference. It fuses the representations of every data-rich class according to their relation strength with the target sample and uses the fusion representation for prediction. The method outperforms the SOTA method in the experiments, and the experiments also demonstrate the interpretability and robustness of the method.

**Strengths:**

1. Sound motivation: the method can automatically learn the relation between the target sample and data-rich classes, which is useful for few-shot learning.
2. Good overall method design: the overall method is well-designed and derived from reasonable mathematical intuitions. The authors get inspiration from human perceptions that perform relation thinking as a sampling of a Bernoulli distribution, and they further approximate it by Gaussian distribution based on the De Moivre-Laplace theorem.
3. Effectiveness: The experimental results outperform the SOTA method on two datasets.
4. Interpretability and robustness: The experiment results demonstrate its interpretability and robustness.

**Weaknesses:**

1. Presentation: I can see the authors have made a lot of revisions to improve the presentation based on other reviewers’ feedback, and I really appreciate it. However, there are still some issues. 1) Some notations are confusing. For example, $\rm N_b$ represents the number of base class nodes, but $N_b$ represents the base class embeddings. It is better to use different letters. There is $\tilde{\sigma}_i$ in Equation 5, but it becomes $\sigma_i$ in Equation 16. 2) The term "Bernoulli distribution" used in the paper is actually the binomial distribution. 3) It needs to take some effort to understand the logic in some paragraphs. 4) A previous paper on distribution calibration [1] provides some visualizations of the distribution that can facilitate understanding (Figure 1, Figure 2, and Figure 6). It would be helpful to provide some visualizations.
2. Lack of experiments on more powerful neural networks: The method is only evaluated on simple models such as logistic regression.

[1] Shuo Yang, Lu Liu, and Min Xu. Free lunch for few-shot learning: Distribution calibration. arXiv preprint arXiv:2101.06395, 2021.

**Questions:**

1. Why does the training phase use a linear classifier, but the inference phase uses a logistic regression classifier?
2. What is the prior graph and posterior graph in Equation 15?
3. How is $m_i$ derived in Equation 5?

---

> ### Author Response · Authors · 2023-11-30
> **Main Responses to Reviewer PCKz (1/2)**
>
> First of all, we would like to thank the reviewer for their encouragement to our work. In this work, we propose a calibration distribution method based on Bayesian relation reference method for few-shot classification task. The highlight of the proposed model is that it **achieves better performance** compared to the SOTA few-shot learning algorithms, while visual analysis proves that the proposed model **has a certain degree of interpretability**, which is useful especially for medical scenarios such as dermatological classification.
>
> Thanks very much for pointing out the weaknesses of the article. It helps us to better improve the deficiencies in our paper. We summarize the weaknesses and the improvements based on these weaknesses as follows:
>
> - ***W1:*** *Some notations are confusing.*
>
>   ***I1:*** Thank you for your constructive comment.  We revised the manuscript, changing notations that were ambiguous to the paper. Specifically, we **replace the notation** "$\rm{N}$", which denotes the dimension, with "$\rm{K}$", and **harmonize the representation** of σ throughout the text. as follows:
>
>   >  $\rm{K_{b}}$ represents the number of base class nodes, $\boldsymbol{N_{t}} \in \mathbb{R}^{\rm{K_{f}}}$ represents the target node embedding,
>
>   >  ${\\sigma}\_{\mathrm{i}}=\operatorname{\zeta}\left(\  \mathrm{L}\_{\mathrm{std}}\left(\boldsymbol{E}\_{\mathrm{e}\_{\mathrm{i}}}\right)\ \right)$
>
>
>
> - ***W2:*** *The term "Bernoulli distribution" used in the paper is actually the binomial distribution.*
>
>   ***I2:*** Thank you for pointing out our clerical error, we have corrected the part of the manuscript that was incorrectly written as "Bernoulli distribution" to "binomial distribution".
>
>
> - ***W3:*** *It needs to take some effort to understand the logic in some paragraphs.*
>
>   ***I3:*** Thank you for your suggestion for our paper. In previous revisions, we have endeavored to make the formulas in the manuscript less difficult to understand, but some of the mathematical transformations used (e.g., the transform from binomial distribution to Gaussian distribution) could not be recounted in detail due to space limitations in the article. We show these theorems in the supplementary material and marked in the **manuscript** (e.g., *Please refer to the supplementary material A.3.2 for the calculation of this value* in Section 3.3, page 6). **In addition we have made some wording changes to improve the readability of the article.**
>
> - ***W4:*** *A previous paper on distribution calibration [1] provides some visualizations of the distribution that can facilitate understanding (Figure 1, Figure 2, and Figure 6). It would be helpful to provide some visualizations.*
>
>   ***I4:*** Thank you very much for your suggestion, the article you mentioned is an inspiring paper and we will refer to it to enrich the content of our manuscript and make it more understandable. Due to time constraints, **these changes will be finalized after the rebuttal and uploaded to the anonymous GitHub**.

---

> > ### Author Response · Authors · 2023-11-30
> > **Main Responses to Reviewer PCKz (2/2)**
> >
> > Thanks very much for the suggestions and questions raised by the reviewer. We answer the questions raised by the reviewer below:
> >
> >
> >
> > - ***Q1:***  *Why does the training phase use a linear classifier, but the inference phase uses a logistic regression classifier?*
> >
> >   **A1:** Thank you for your constructive question. In the paper, the training process is modeled after conventional deep learning with the aim of optimizing the Bayesian relation inference component. Due to the **high number of classes** in training data (more than 200 classes), the **logistic regression classifier is not as effective as the linear classifier** for this type of problem. Whereas only five-class tasks are required during validation and testing, logistic regression classifier is faster to train. Moreover, existing work (Yang et.al. 2021) indicates that the logistic regression classifier outperforms many other classifiers in distributional calibration tasks.
> >
> > - ***Q2:***  *What is the prior graph and posterior graph in Equation 15?*
> >
> >   **A2:** The edges of the post graph are $m_{\mathrm{i}}$, and the prior graph is obtained by fitting the mean and variance of the Gaussian distribution of its edges and sampling them using a linear neural network.
> >
> > - ***Q3:*** *How is $m_{\mathrm{i}}$ derived in Equation 5?*
> >
> >   **A3:** We assume that the summary graph of the coupling is a sampling of a binomial distribution $\mathcal{B}(\,n, \lambda\,)$ with $n\rightarrow \infty$ and $\lambda \rightarrow 0$.} However, due to the $n\rightarrow \infty$ of this binomial distribution, we **cannot directly solve for the specific values of the edges** of the summary graph. By **theorem 1 shown in supplementary material A.3.1**, we can use the method shown in the manuscript to derive $m_{\mathrm{i}}$. Specifically, $m_{\mathrm{i}}$ in the summary graph is **sampling of Gaussian proxies for the hypothetical binomial distribution** being used to **compute the KL term** and, by Theorem 1, we are assured that the **error of the computation using this $m_{\mathrm{i}}$ is acceptable.** The following is the definition for $m$ in Theorem 1:
> >
> >   > according to exist works, we have that: $f_{1}(x)$ attains its minimum on the interval $(0,1)$ and $f_{2}(x)-f_{2}^{*}$ is bounded on the interval $(0, \sqrt{2} / 2-   1 / 2)$, with:
> >   >
> >   > $x=m=\frac{1+l-\sqrt{1+l^{2}}}{2}, \text { where } l=\frac{2 \sigma^{2}}{1-2 \mu}$
> >
> > Thank you for your patience in reading through our responses. Your constructive comments further improve the quality of our paper.**We sincerely hope that the revised paper can receive a higher score from you, thank you!**

---

> ### Comment · Reviewer_PCKz · 2023-12-03
> **Further Questions**
>
> Thank you for your response! I have a clearer understanding, but I am still confused about a few issues.
>
> 1. I am still confused about how $m_i$ is derived. By comparing the theorem and Equation 5, it seems that $l = 2 \sigma_i^2 / (1 - 2 \mu) = 2 n_i \sigma_i^2$, which leads to $n_i = \frac{1}{1-2\mu}$. How is it obtained?
> 2. The writing still needs to be clearer, for example, the A.1.3 subsection in the appendix should explain what the hyper-parameters are.
> 3. Can you specifically describe how you optimize the model? Do you train the classification head and the Bayesian relational inference component together or sequentially? During the validation and test phase, I guess you fix the Bayesian relational inference component and simply optimize the logistic regression classifier, is my understanding correct? If you train the logistic regression classifier on the validation or test set, is there data leakage, and do the baseline methods also optimize some parts of the neural network on the validation and test set?
> 4. You claimed in the conclusion that "to the best of our knowledge, this is the first attempt to combine the Bayesian relational inference method with distribution calibration." One of the main novelties you state is that the proposed method can automatically calculate the relation score, while previous methods usually used manually designed distance measurements. Is there any previous method that also calculates the relation automatically? If there are some, could you please explain your uniqueness?

---

### Meta-Review · Area_Chair_h2yk · 2023-12-22

**Metareview:**

The paper proposes a novel distribution calibration method for few-shot learning based on Bayesian relation inference.

All reviewers agreed that the proposed method is novel and interesting. The authors have addressed some of my concerns during the rebuttal, but I think the presentation still needs to be improved, and adding more experiments can lead to more contributions, in my opinion. If I can be of any further help, please do not hesitate to contact me. Thank you very much!

However, all reviewers also agreed that the presentation of the paper should be improved and that more experiments should be added to improve the experimental analysis.
The authors did a good job of improving the paper in the rebuttal phase but during the reviewer discussion, there was the consensus that the issues could not be fully solved.

In addition, I missed some important information in the main paper, like. how many views per target were considered, and ---since the few-shot classification results depend on the random graphs beeing generated --- information on how results vary with randomness (i.e. mean and variance of performance for several runs should be reported.  This seems esspecially important, since the rerported performance differences to baselines are not big.

**Justification For Why Not Higher Score:**

Presentation and experiments could be improved

**Justification For Why Not Lower Score:**

N/A

---

### Decision · Program_Chairs · 2024-01-16

Reject